# The amount of membrane cholesterol required for robust cell adhesion and proliferation in serum-free condition

Shino Takii[1], Jun Wu[2,3], Daiji Okamura[1] *

**1** Department of Advanced Bioscience, Graduate School of Agriculture, Kindai University, Nara, Japan,
**2** Department of Molecular Biology, University of Texas Southwestern Medical Center, Dallas, TX, United States of America, **3** Hamon Center for Regenerative Science and Medicine, University of Texas Southwestern Medical Center, Dallas, TX, United States of America

* dokamura@nara.kindai.ac.jp

**Data Availability Statement:** All relevant data are contained within the paper.

**Funding:** Financial Disclosure: This work was supported by grants from JSPS KAKENHI Grant

## Abstract

Serum-containing medium is widely used to support cell attachment, stable growth and serial passaging of various cancer cell lines. However, the presence of cholesterols and lipids in serum greatly hinders the analysis of the effects of cholesterol depletion on cells in culture. In this study, we developed a defined serum-free culture condition accessible to a variety of different types of adherent cancer cells. We tested different factors that are considered essential for cell culture and various extracellular matrix for plate coating, and found cells cultured in Dulbecco's Modified Eagle's Medium (DMEM) basal media supplemented with Albumin (BSA) and insulin-transferrin-selenium-ethanolamine (ITS-X) on fibronectin-precoated plate (called as "DA-X condition") showed comparable proliferation and survival to those in a serum-containing medium. Interestingly, we observed that DA-X condition could be adapted to a wide variety of adherent cancer cell lines, which enabled the analysis of how cholesterol depletion affected cancer cells in culture. Mechanistically, we found the beneficial effects of the DA-X condition in part can be attributed to the appropriate level of membrane cholesterol, and fibronectin-mediated signaling plays an important role in the suppression of cholesterol production.

## Introduction

Modern advances in medical and biological sciences have largely relied on the development of cell culture technology [1]. The ability and quality of culture medium to support cell survival, proliferation, and function in vitro have a direct impact on research outcomes. Thus, it is essential to select the appropriate medium when conducting cell culture experiments. It is well-established that serum-containing medium provides an optimal culture condition, which is widely used to support attachment, stable growth and serial passaging of various cancer cell lines in culture. However, as cell culture research progressed, the need for serum-free culture media, which are expected to help overcome various ethical and scientific issues, became apparent [2]. Compared to serum-containing media, serum-free media have advantages such

Number JP20K06661 (Grant-in-Aid for Scientific Research (C)) and Kindai University Research Enhancement Grant(KD2004 and KD2101). The funders had no role in study design, data collection and analysis, decision to publish, or preparation of the manuscript.

**Competing interests:** The authors have declared that no competing interests exist.

as less variability between lots, more consistent, and in many cases lower cost (unless expensive growth factors and cytokines are used) [3].

Cholesterol, an essential component of mammalian cell membranes, not only maintains cell structure, but also plays an important role in other cellular functions such as biosynthesis of bile acids and hormones, embryonic development, and cell proliferation [4]. Most of the cellular cholesterol enriches in the plasma membrane after transportation from endoplasmic reticulum, which regulates cellular proliferation, differentiation, and survival. Cholesterol metabolism is known to critically contributes to cancer cell proliferation, migration and invasion, and accumulation of cholesterol in solid tumors is considered as a hallmark for aggressive cancers [5–9]. However, given that serum contains numerous cholesterols and lipids, serum-containing media are therefore not ideal for studying effects of cholesterols on cancer cells in culture. Although serum-free media with optimized compositions for each adherent cancer cell line are available, few of these cultures can support the growth of other cancer cell types, and there is a lack of universal serum-free medium applicable to a wide range of adherent cell lines. In contrast, in the field of pluripotent stem cells, the use of serum-free and/or xeno-free medium has become standard [10].

In this study, we aimed to develop a universal serum-free culture medium broadly applicable to any adherent cancer cell types. While testing various essential culture parameters including extracellular matrix, we found that growing cells in Dulbecco's Modified Eagle's Medium (DMEM) basal medium containing Albumin (BSA) and insulin-transferrin-selenium-ethanolamine (ITS-X) on culture plates pre-coated with fibronectin (called as "DA-X condition") showed robust cell proliferation and attached cells exhibited elongated pseudopodia, which were comparable to cells grown in a serum-containing medium. DA-X condition also facilitated the study of the effects of cholesterol and lipid on cancer cells in vitro in a wide variety of adherent cancer cell lines. Importantly, we found that cells grown in the DA-X condition maintained the right amount of membrane cholesterol important for robust cell attachment, elongated pseudopodia and survival in serum-free condition, and fibronectin-mediated signaling plays important roles in suppressing excess cholesterol production.

## Materials and methods

### Cell lines and culture

Cell lines using in this research were obtained from the Cell Resource Center for Biomedical Research (Institute of Development, Aging and Cancer, Tohoku University, Japan) and maintained in 10% Fetal Bovine Serum (Gibco, 10270)-contained DMEM medium (nacalai tesque, 08458–45) which is supplemented with 1x Penicillin-Streptomycin Mixed Solution (nacalai tesque, 26253–84), and passaged using TrypLE (Gibco, 12604013) every 4–5 days. Briefly for in RPMI-G [11], $2x10^4$ cells were seeded into one well of a 4-well plate without pre-coating in RPMI1640 medium (nacalai tesque, 3026485) supplemented with 1x ITS-G: Insulin-Transferrin-Selenium (Gibco, 41400045), 1x L-Glutamine (nacalai tesque, 16948–04), 1x Penicillin-Streptomycin Mixed Solution. In DA-X condition (shown as ITS-X/ DMEM/ BSA (FN) in Fig 1A), $2x10^4$ cells were seeded into one well of a 4-well plate pre-coated with Fibronectin in DMEM medium (High Glucose) (nacalai tesque, 08458–45) supplemented with 1x ITS-X: Insulin-Transferrin-Selenium-Ethanolamine (Gibco, 51500056), 5 mg/mL BSA: Bovine Serum Albumin (Sigma, A3059), 1x L-Glutamine, 1x Penicillin-Streptomycin Mixed Solution. DMEM/Ham's F-12 (nacalai tesque, 11581–15) was used for ITS-X/ DMEM/F-12/ BSA (FN) (shown in Fig 1B). Extracellular matrix was coated on a well of 4-well cell culture plate (SPL, 30004) and incubated for 1 hour at 37˚C at the desired concentration, 2.35 μg/cm² for Fibronectin (FUJIFILM, 063–05591), 1.25 μg/cm² for Vitronectin (FUJIFILM, 220–02041), 0.5 μg/

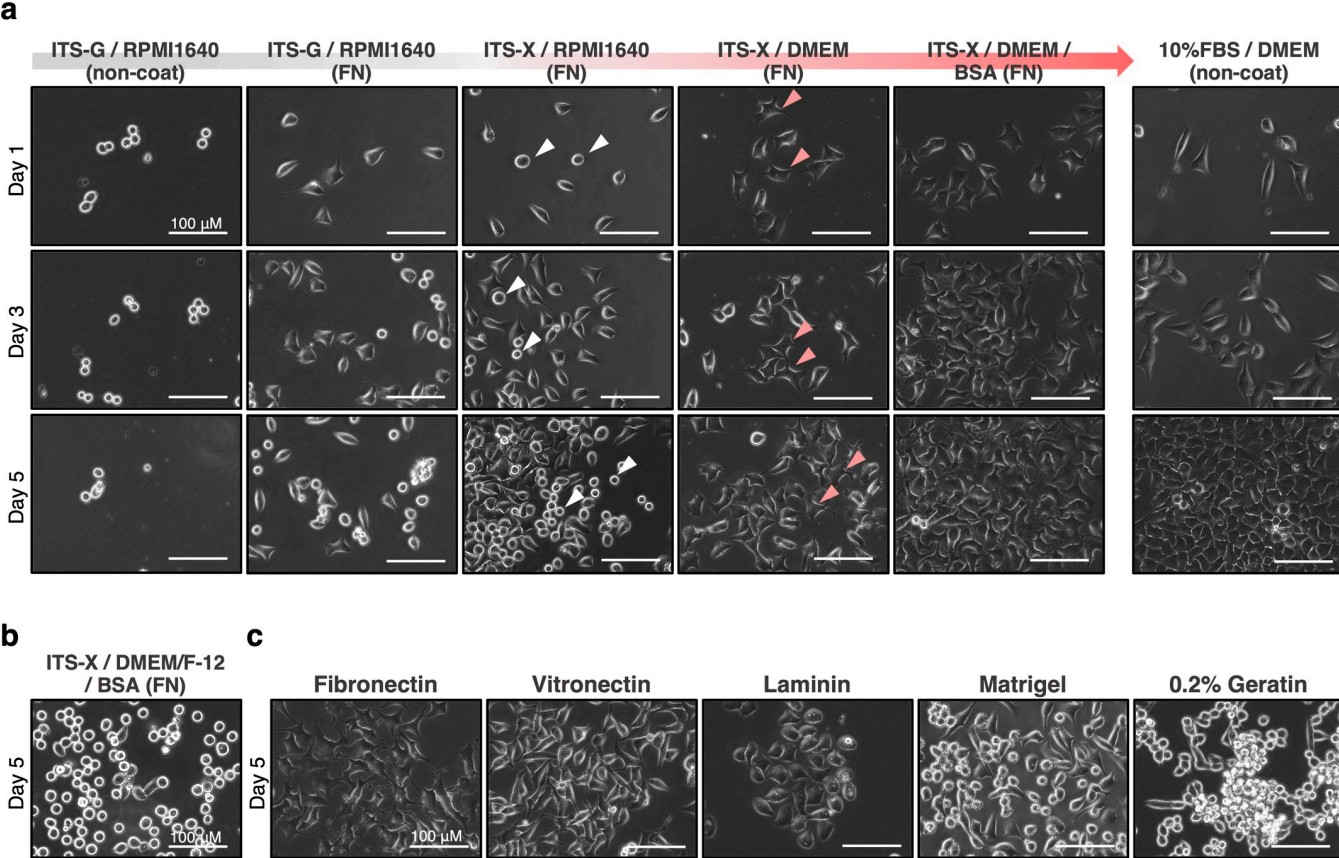

**Fig 1. Optimization of novel serum-free condition on HeLa cells. a,** Representative images of cells in each culture condition. In screening for novel serum-free condition, $2x10^4$ cells of HeLa cell (human cervical cancer cell line) maintained in 10% FBS/ DMEM were seeded onto each serum-free media condition. Combinations of media composition and extra-cellular matrix are shown. Improving stepwise from RPMI-G with replacing the media composition resulted in establishment of the ITS-X/ DMEM/ BSA on fibronectin-precoated well (called as DA-X condition), which significantly improved cell adhesion, pseudopodia elongation, or proliferation. Non-coat, without any pre-coating. White and pink arrowheads indicate cells with losing attachment and cells with firm elongated pseudopodia, respectively. FN, fibronectin-coated. Scale bar, 100 μm. **b,** Representative images of cells cultured in DMEM/ Ham's F-12-based medium with ITS-X/ BSA on fibronectin-precoated well. **c,** Representative images of cells in comparison of effect of extracellular matrix. $2x10^4$ cells of HeLa cell maintained in 10% FBS/ DMEM were seeded onto a pre-coated well with each extracellular matrix. Regarding to cell adhesion, pseudopodia elongation, or proliferation, fibronectin was shown as much better than the other extracellular matrix. The effect of vitronectin was shown as comparable to that of fibronectin in cell adhesion and cell proliferation, but not in pseudopodia elongation. Scale bar, 100 μm.

$cm^2$ for Laminin511-E8 fragments: iMatrix-511 (MAX, 892011), 2% for Matrigel (Corning, 354234), 0.2% for Gelatin (Sigma, G1890). For optimization of serum-free culture condition for stable adhesion, the combination of basal medium, supplements and extracellular matrix shown in the notation in Fig 1A at a concentration mentioned above were attempted. For inhibition of fibronectin-binding and -mediated signaling, RGDS peptide (Cayman, 15359–5) as fibronectin Inhibitor was added in culture at 50 μg/mL (the solvent, DMSO, was added for control). The images of cells were collected by a microscopy in phase-contrast (Keyence, BZ-X710).

## Modulation of cholesterol

For modulating of membrane cholesterol contents in culture, Methyl-β-cyclodextrin (MβCD) (Sigma, 332615) for depletion of membrane cholesterol [12] and soluble cholesterol (Sigma, C8667) (as a complex of MβCD and cholesterol) were added according to each purpose at 0.2 mM, 1 mM (MβCD) and 30 μM (cholesterol). $2x10^4$ cells maintained in 10% FBS/ DMEM

were seeded onto non-coat in RPMI-G or fibronectin-coated in DA-X condition well of 4-well cell culture plate. At 1 day after seeded, MβCD or soluble cholesterol was added (the solvent, sterile water or EtOH, was added for control respectively). For exploring of functions of cholesterol biosynthesis depending on culture medium, Ro48-8071 (Cayman, 10006415), a selective inhibitor of Oxidosqualene cyclase [13], known as a cholesterol biosynthesis inhibitor was used. $2 \times 10^4$ cells maintained in 10% FBS/ DMEM were seeded onto non-coat in 10% FBS/ DMEM or fibronectin-coated in DA-X condition well of 4-well cell culture plate. At 1 day after seeded, Ro48-8071 was added at 1 μM (the solvent, EtOH, was added for control).

## Qualitative estimation of membrane cholesterol

For labeling of membrane cholesterol, cells grown on 4-well plate were fixed with 4% paraformaldehyde in PBS for 15 min at room temperature. The cells were washed three times in PBS, and then incubate with 1.5 mg/ml glycine in PBS for 10 min at room temperature to quench the paraformaldehyde. Cells were stained with filipin working solution (0.05 mg/ml filipin III (Cayman, 70440) in 10% FBS-contained PBS) for 2 hrs at room temperature, and washed three times in PBS, and then the images were collected by fluorescent microscopy in PBS in phase-contrast (Keyence, BZ-X710).

## RNA preparation and real-time PCR

Total RNAs were extracted by using Sepasol-RNA I Super G (nacalai tesque, 09379) to the manufacturer's instructions. RNAs were reverse-transcribed using ReverTra Ace qPCR RT Master Mix (TOYOBO, FSQ-201), and real-time PCR was performed using THUNDERBIRD SYBR qPCR Mix (TOYOBO, QPS-201) in MIC qPCR (bio molecular systems). Expression levels of each gene were normalized to β-ACTIN (human) expression and calculated using comparative CT method. The primer sequences are shown in below: β-ACTIN-F (`CTGGCA CCACACCTTCTACAATG`), β-ACTIN-R (`AATGTCACGCACGATTTCCCGC`), SREBF1-F (`ACAGTGACTTCCCTGGCCTAT`), SREBF1-R (`GCATGGACGGGTACATCTTCAA`), SREBF2-F (`AACGGTCATTCACCCAGGTC`), SREBF2-R (`GGCTGAAGAATAGGAGTTGCC`), ACSS2-F (`AAAGGAGCAACTACCAACATCTG`), ACSS2-R (`GCTGAACTGACACACTTGGAC`), HMGCR-F (`TGATTGACCTTTCCAGAGCAAG`), HMGCR-R (`CTAAAATTGCCATTCCACGAGC`), HM-GCS1-F (`GATGTGGGAATTGTTGCCCTT`), HMGCS1-R (`ATTGTCTCTGTTCCAACTTCC AG`), LDLR-F (`ACCAACGAATGCTTGGACAAC`), LDLR-R (`ACAGGCACTCGTAGCCGAT`), ACLY-F (`TCGGCCAAGGCAATTTCAGAG`), ACLY-R (`CGAGCATACTTGAACCGATTCT`).

## Statistical analysis

Statistical analysis was performed using the Student's t-test. P values < 0.05 were considered to be statistically significant.

## Results

### Development of a novel serum-free culture condition for HeLa cells

ITS-G/ RPMI1640 (referred to as "RPMI-G" hereafter), a defined serum-free cell culture medium reported previously [11], contains ITS-G supplement composed of Insulin-Transfer-rin-Selenium, known to support cell proliferation in reduced-serum medium. RPMI-G supported robust cell adhesion and proliferation of several melanoma cell lines. We tested culturing HeLa cells in RPMI-G medium. Interestingly, however, HeLa cells loosely attached to the culture plate 2~3 days after seeding and detached, and consequently could not be maintained in RPMI-G medium (Fig 1A). To establish a simpler and more reliable serum-free

condition that can facilitate the functional analysis of cholesterol and lipid metabolism of various adherent cancer cell lines including HeLa cells, we optimized the RPMI-G medium taking consideration of several culture parameters: (1) Cell adhesion to culture plates, (2) Cell morphology (with or without pseudopodia), and (3) Cell proliferation. We didn't consider the long-term cultivability in this report. After testing the effects of various supplements, base medium and extracellular matrix in serum-free conditions, we found ethanolamine in ITS-X, fibronectin-precoating and DMEM base medium had positive effects on cell adhesion and extension of pseudopodia, and the supplementation of BSA improved cell proliferation (Fig 1A). Interestingly, we found although DMEM/Ham's F-12 is widely used as a basal media for serum-free cultures [1], cell attachment was markedly attenuated with the retraction of pseudopodia on day 5 while initial cell attachment and growth were not affected (Fig 1B).We also tested the effects of several extracellular matrix proteins, which are widely used in pluripotent stem cells including human iPS cells (Fig 1C) [1, 2]. Taken together, ITS-X/ DMEM/ BSA on fibronectin pre-coated culture plates, referred to as the "DA-X condition", was determined to be the optimal serum-free culture condition that supports stable adhesion, extended pseudopodia and robust cell proliferation, which is comparable to 10% FBS/ DMEM medium condition (Fig 1).

## The utility of DA-X condition for studying cholesterol function

It is difficult to see the early effects of cholesterol biosynthesis inhibition or depletion of membrane cholesterol (e.g. treatment with Methyl-β-cyclodextrin [12]) in adherent cultures, since serum even at reduced levels contains large amounts of lipids and cholesterols. In fact, Ro48-8071 known as a cholesterol biosynthesis inhibitor by selective inhibition of Oxidosqualene cyclase had no harmful effects at all even at 1 μM on adherent HeLa cells cultured in 10% FBS/ DMEM medium (Fig 2A, two images at the top). In sharp contrast, addition of Ro48-8071 in DA-X condition had a dramatic effect, and almost all cells detached and underwent cell death within 48 hours (Fig 2A, two images at the bottom). These results demonstrate the potential of the serum-free DA-X condition to provide an appropriate culture environment for analyzing the role of lipids and cholesterols in adherent cancer cell lines.

## DA-X condition as universal serum-free condition for cancer cell lines

The development of serum-free media that can be broadly applied to any cancer cell line is important for studying the role of lipids and cholesterols in the survival and proliferation of cancer cells. Next, we performed side-by-side comparison of RPMI-G medium and DA-X condition in terms of cell adhesion, pseudopodia and proliferation using representative human cancer cell lines derived from different organs and tissues (Fig 2B and 2C). Cancer cells maintained in 10% FBS/ DMEM medium were switched to culture in serum-free conditions RPMI-G and DA-X, and phase contrast images were recorded on day 5 to monitor cell viability, attachment, morphology, and proliferation. In general, we found DA-X condition was more reliable than RPMI-G to support cell survival and proliferation with pronounced elongated pseudopodia in tested cell lines (Fig 2B and 2C). Cell proliferation and attachment of SH-SY5Y (neuroblastoma) and PC-3 (prostatic cancer) in DA-X condition was comparable to those in RPMI-G. Although HepG2 (hepatoma) and SW620 (colon cancer) in DA-X condition showed comparable proliferation when compared to that in RPMI-G, they exhibited more evident adhesion and pseudopodia in cells than those in RPMI-G. In the other lines including A549 (lung cancer), Panc-1 (pancreatic cancer), HeLa (cervical cancer), MCF-7 (breast cancer), cell survival and proliferation were observed only under DA-X condition but not in RPMI-G. Taken together, we demonstrate the efficacy in culturing various adherent cancer

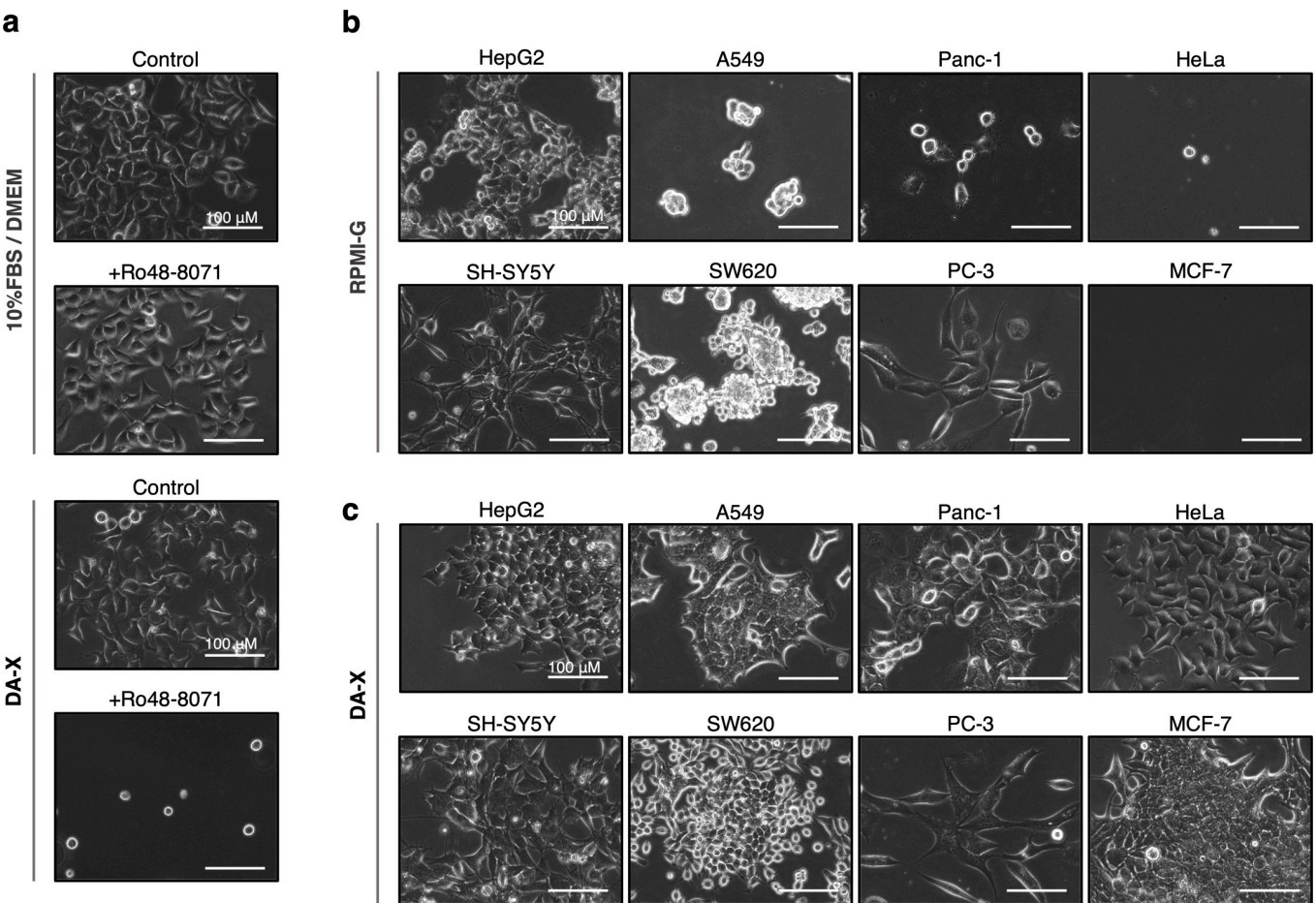

**Fig 2. Availability of DA-X condition on analysis for cholesterol function, and in several cancer lines. a,** Effect of cholesterol biosynthesis inhibitor, Ro48-8071, on HeLa cells in serum-containing and -free culture condition for 48 hrs. In cells grown in 10% FBS/ DMEM, Ro48-8071 was shown to have no effect on cell survival and even their proliferation, while almost cells cultured in DA-X condition were shown to be dead. Scale bar, 100 μm. **b** and **c,** Comparison of the effects of RPMI-G medium and DA-X condition on cell adhesion, pseudopodia and proliferation in representative human cancer cell lines of each organ and tissue. Images of cells cultured in RPMI-G (b) and DA-X condition (c) at day 5 are shown. The number of cells at the start of culture was optimized for each cell type to reach about 80% confluency at day 5 in 10% FBS/ DMEM medium (the using cancer cell lines and their number of cells at the start of cultures in one well of a 4-well plate are shown below). HepG2 (hepatoma, $4\times10^4$), A549 (lung cancer, $2\times10^4$), Panc-1 (pancreatic cancer, $4\times10^4$), HeLa (cervical cancer, $2\times10^4$), SH-SY5Y (neuroblastoma, $4\times10^4$), SW620 (colon cancer, $4\times10^4$), PC-3 (prostatic cancer, $3\times10^4$), MCF-7 (breast cancer, $3\times10^4$). Scale bar, 100 μm.

cell lines in the serum-free DA-X condition (at least for 5 days without passaging), which holds great potential for analyzing role of lipids and cholesterol in these cells (Fig 2).

## Varied cholesterol content in serum-free culture medium condition

RPMI-G, a defined serum free cell culture medium, supports de novo fatty acid and -cholesterol biosynthesis in several adherent melanoma cell lines, which enabled unperturbed cell adhesion and proliferation [11]. While testing the hypothesis that cells grown in the DA-X condition have increased fatty acid and cholesterol production through upregulation of biosynthesis-related genes when compared to the RPMI-G culture medium, we surprisingly found that the amount of cholesterol in the membrane of HeLa cells in DA-X medium was greatly reduced in comparison to that in RPMI-G (Fig 3A). Consistently, significant transcriptional down-regulation of cholesterol biosynthesis-related genes was also observed in cells cultured in the DA-X condition. In contrast, these genes were up-regulated in RPMI-G when

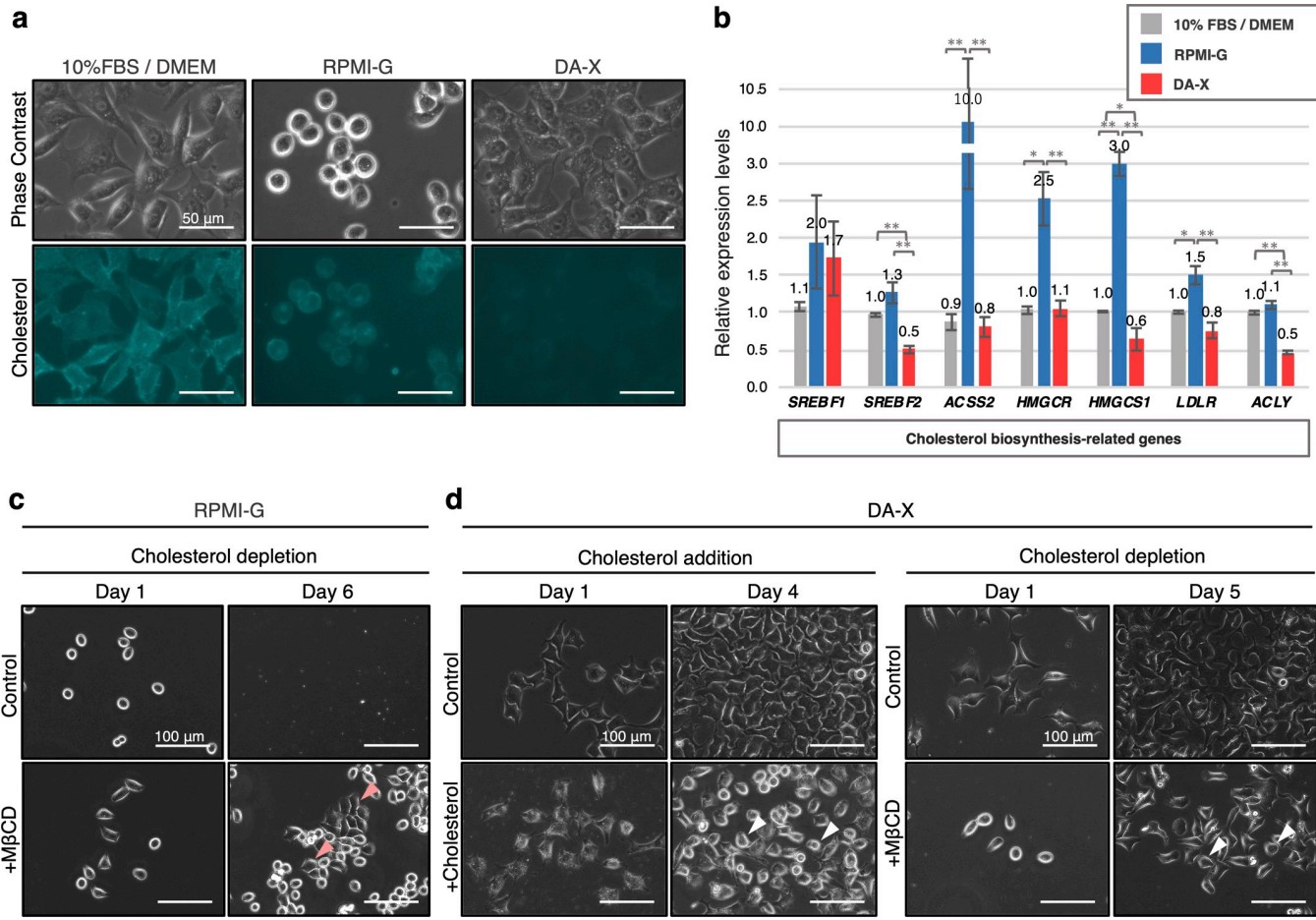

**Fig 3. Required cholesterol level on membrane optimized by DA-X condition in HeLa cells. a,** Cholesterol labeling with filipin fluorescence staining in each culture condition. Cells were cultured in 10% FBS/ DMEM, RPMI-G, DA-X condition for 2 days, and then applied for the staining. Signals of fluorescence were enriched in membrane of cells regardless of culture condition, and the intensity was strong in 10% FBS/ DMEM and RPMI-G, but quite faint in DA-X condition. Images for filipin staining were obtained using the same exposure time. Scale bar, 50 μm. **b,** Quantitative PCR analysis of expression of de novo cholesterol biosynthesis-related genes. Cells cultured in each condition for 2 days, and collected for further gene expression analysis. Error bars indicate s.d. (n = 3, biological replicates). t-test, $**p < 0.01$ and $*p < 0.05$. **c,** Enhancement of cell adhesion and proliferation by depletion of membrane cholesterol with Methyl-β-cyclodextrin (MβCD) at 0.2 mM in RPMI-G culture medium. Pink arrowheads indicate cells with firm elongated pseudopodia. **d,** Effects of excess and depletion of membrane cholesterol on cell adhesion by addition of soluble cholesterol and MβCD in DA-X condition. At day 4 and 5, attenuated cell adhesion with the retraction of pseudopodia was widely observed not only in addition of cholesterol, but also in depletion of membrane cholesterol with MβCD at 1 mM. Scale bar, 100 μm. White arrowheads indicate cells with losing attachment.

compared with 10% FBS/ DMEM as previously reported (Fig 3B) [11]. Together, these results lead us to hypothesize that cells grown in RPMI-G medium are prone to detach due to the excess of membrane cholesterol, DA-X condition provides benefit for stable cell adhesion in part due to lowered membrane cholesterol content.

## Range of optimal membrane cholesterol for stable cell adhesion

To determine whether there is an appropriate range of membrane cholesterol for stable cell adhesion in serum-free medium condition, we modulated the membrane cholesterol levels in both RPMI-G and DA-X conditions to see their effects on cell adhesion. Augmentation and depletion of membrane cholesterol were achieved by supplementation with soluble cholesterol and Methyl-β-cyclodextrin (referred to as "MβCD" hereafter), respectively (Fig 3C and 3D). Depletion of cellular membrane cholesterol by MβCD in RPMI-G (Fig 3A and 3B) resulted in

a significant improvement in cell adhesion (Fig 3C). In contrast, under DA-X conditions, both increase and depletion of membrane cholesterol led to the attenuation of cell adhesion accompanied by the retraction of elongated pseudopodia (Fig 3D). These results suggest that the excess of membrane cholesterol in RPMI 1640 medium may have caused cell death due to compromised cell adhesion, and the beneficial effects of the DA-X condition may due to the appropriate amount of membrane cholesterol achieved by the suppression of cholesterol biosynthesis-related gene expression (Fig 3) [14].

## Involvement of fibronectin in the suppressing genes needed for cholesterol biosynthesis

To determine the molecular mechanisms by which expression of cholesterol biosynthesis-related genes were suppressed in DA-X condition, we inhibited fibronectin-binding and -mediated signaling by the addition of RGDS peptide [2, 15, 16], and quantified the expression levels of cholesterol biosynthesis-related genes by quantitative PCR analysis (Fig 4A and 4B). Inhibition of fibronectin with RGDS peptide in DA-X condition resulted in not only different cell morphology (Fig 4A) but also significant up-regulation of cholesterol biosynthesis-related genes (Fig 4B). Therefore, fibronectin-binding or -mediated signaling play important roles in the suppression of cholesterol production in DA-X condition.

## Discussion

Here, we show that the amount of cholesterol in the cell membrane greatly affects cell attachment, pseudopodia formation and eventually the survival of cells under serum-free conditions. Furthermore, functional analysis by modulating membrane cholesterol content in RPMI-G and DA-X conditions revealed that there is the appropriate amount of range of membrane cholesterol for stable cell attachment, pseudopodia formation and proliferation in serum-free medium condition (Figs 3C, 3D and 4C). We demonstrated that cells grown in the DA-X culture condition maintained the right amount of cholesterol in cell membrane in part through extracellular matrix (fibronectin) signaling. The positive results seen using eight representative human cancer cell lines suggest that the DA-X condition serves as a universal serum-free culture condition that support pseudopodia formation, robust cell attachment and proliferation prior to cell passaging. This contrasts with conventional serum-free media, which are typically developed and optimized for each cell types. We therefore believe the DA-X culture condition can serve as a powerful platform for clarifying the role of cholesterol in different cancer cells [1].

We also demonstrated that supplementation of Ro48-8071 in DA-X condition had profound effects on cell detachment and viability (Fig 2A). Statin, inhibitors of HMG-CoA reductase, has also shown to inhibit cancer cell proliferation and increase apoptosis [8, 17, 18], which seems dependent on cell attachment since similar effects were not observed in acute lymphoid leukemia grown in suspension [7]. Future studies are warrantied to further examine the mechanism(s) of cell death by cholesterol biosynthesis inhibition on cancer cells.

We observed that marked increase of membrane cholesterol content in HeLa cells cultured in RPMI-G (Fig 3A and 3B), which is consistent with seen in other type of serum-free conditions [19]. Cells in these conditions are prone to detachment and cell apoptosis. Enrichment of cholesterol in the lipid raft of membrane in cancer cells make them more sensitive to cholesterol depletion and induces anoikis like cell death [20], suggesting that there is a close relationship between the amount of membrane cholesterol in cancer cells and cell survival in serum-free culture condition.

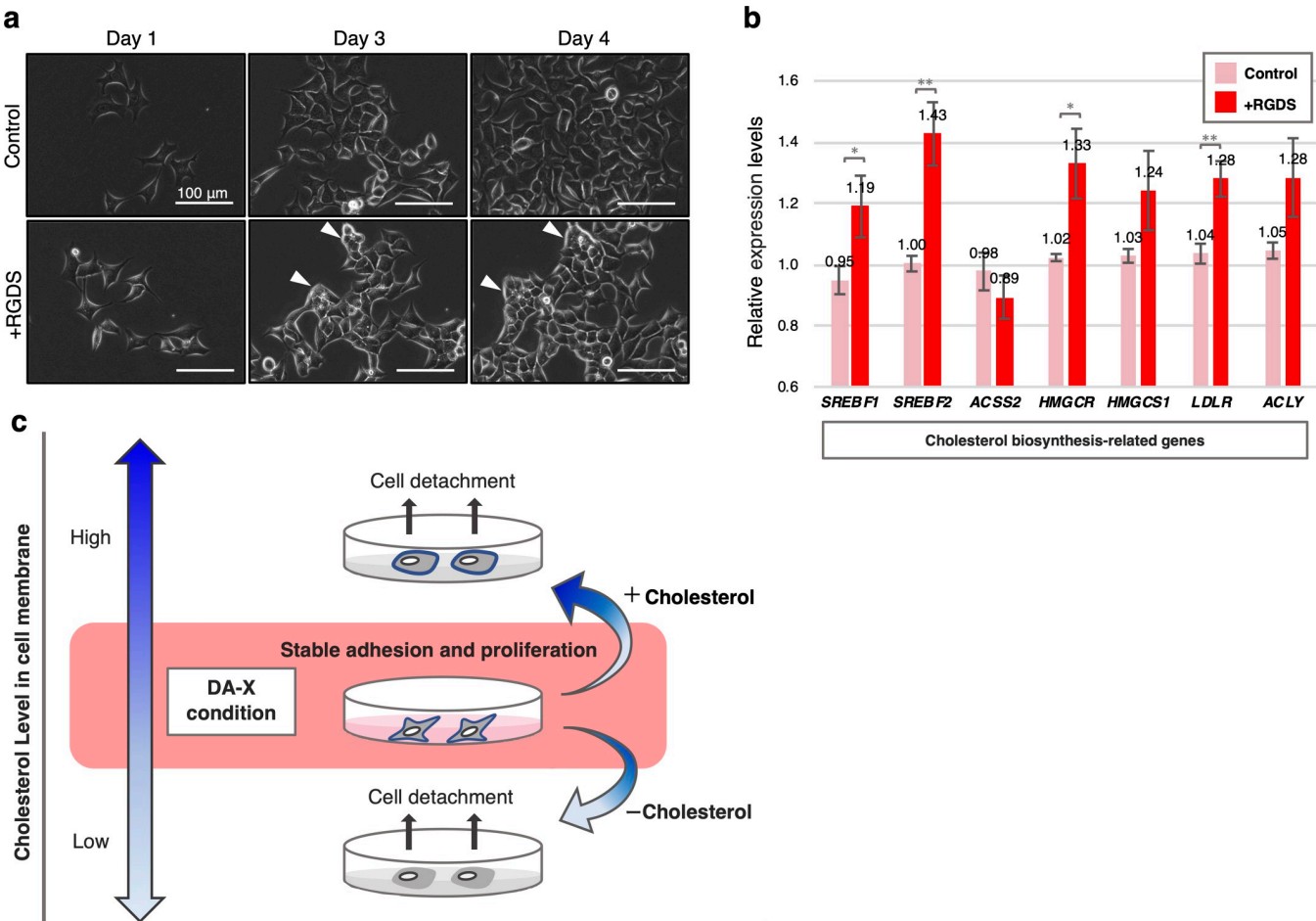

**Fig 4. Involving fibronectin in down-regulation of cholesterol biosynthesis-related genes expression in DA-X condition. a,** Representative images of HeLa cells under RGDS peptide treatment in DA-X culture condition. After exposed to RGDS peptide for 3 days, retracting the pseudopodia of cells was observed accompanied by tightly association between neighbor cells. Scale bar, 100 μm. White arrowheads indicate cells with the retraction of pseudopodia. **b,** Quantitative PCR analysis of expression of de novo cholesterol biosynthesis-related genes. HeLa cells exposed to RGDS peptide for 4 days under DA-X condition were collected for further gene expression analysis. Error bars indicate s.d. (n = 3, biological replicates). t-test, $^{**}p < 0.01$ and $^{*}p < 0.05$. **c,** Schematic representation summarizing the appropriate range of cholesterol contents on cell membrane in serum-free medium condition for cell adhesion and proliferation.

Finally, from the perspective of decarbonization, recently research on cultured meat is rapidly progressing but most, if not all, cultures still use serum-supplemented medium. Serum-free medium will be essential for the establishment of cultured meat in the future in order to reduce cost, achieve consistency and minimize the impact of carbonization [21]. Our study revealed that keeping the right amount of membrane cholesterol is essential for cell survival and proliferation in serum-free culture conditions, and we believe these results will facilitate the development of serum-free medium for cultured meat.

## Supporting information

**S1 Table. Cq value in quantitative PCR analysis as raw data for the gragh.** Cq value for Fig 3B (upper) and Fig 4B (bottom). The expression of cholesterol biosynthesis-related genes as genes of interest is measured between cultured cells in each condition as biological groups. Each biological group contains three biological replicates. *b-ACTIN* is used as a reference gene. (PDF)

## Acknowledgments

We would like to thank Prof. Yoko Kato (Kindai Univ.) and Prof. Yasuhisa Matsui (Tohoku Univ.) for their invaluable comments.

## Author Contributions

**Conceptualization:** Shino Takii, Daiji Okamura.

**Funding acquisition:** Daiji Okamura.

**Investigation:** Shino Takii.

**Supervision:** Daiji Okamura.

**Writing – original draft:** Shino Takii, Daiji Okamura.

**Writing – review & editing:** Jun Wu, Daiji Okamura.

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
