## [Decision Letter · Decision Letter 0]

25 Feb 2022

PONE-D-21-33428The amount range of membrane cholesterol required for robust cell adhesion and proliferation in serum-free condition.PLOS ONE

Dear Dr. Okamura,

Thank you for submitting your manuscript to PLOS ONE. After careful consideration, we feel that it has merit but does not fully meet PLOS ONE’s publication criteria as it currently stands. Therefore, we invite you to submit a revised version of the manuscript that addresses the points raised during the review process.

We look forward to receiving your revised manuscript.

Kind regards,

Avaniyapuram Kannan Murugan, M.Phil., Ph.D.

Academic Editor

PLOS ONE

Journal Requirements:

hen submitting your revision, we need you to address these additional requirements.

3. Thank you for stating the following in the Acknowledgments Section of your manuscript: "This work was partly supported by grants from JSPS KAKENHI Grant Number JP20K06661 (Grant-in-Aid for Scientific Research (C)), Kindai University Research Enhancement Grant (21st Century Joint Research Enhancement Grant), the Cooperative Research Project Program of Joint Usage/Research Center at the Institute of Development, Aging and Cancer, Tohoku University."

Please remove any funding-related text from the manuscript and let us know how you would like to update your Funding Statement. Currently, your Funding Statement reads as follows: "No"

Additional Editor Comments:

Interesting study and reviewers' comments reflected the same. However, some important points raised by the reviewers to be addressed before the publication.

Reviewers' comments:

Reviewer's Responses to Questions

**Comments to the Author**

1. Is the manuscript technically sound, and do the data support the conclusions?

Reviewer #1: Yes

Reviewer #2: Yes

2. Has the statistical analysis been performed appropriately and rigorously? 

Reviewer #1: Yes

Reviewer #2: Yes

3. Have the authors made all data underlying the findings in their manuscript fully available?

Reviewer #1: Yes

Reviewer #2: Yes

4. Is the manuscript presented in an intelligible fashion and written in standard English?

Reviewer #1: Yes

Reviewer #2: Yes

5. Review Comments to the Author

Reviewer #1: Takii et al. submitted a research article entitle “The amount range of membrane cholesterol required for robust cell adhesion and proliferation in serum-free condition”. They aimed to develop a universal serum-free culture medium to facilitate the study of the effects of cholesterol and lipid on cancer cells in vitro in various adherent cancer cell lines. Since the serum-containing media contain numerous cholesterols and lipids, and are not ideal for studying effects of cholesterols on cancer cells in culture, they found that Dulbecco's Modified Eagle's Medium (DMEM) basal medium containing Albumin (BSA) and insulin-transferrin-selenium-ethanolamine (ITS-X) on culture plates pre-coated with fibronectin (called as “DA-X condition”) are comparable to a serum-containing medium. They both showed similar robust cell proliferation and elongated pseudopodia cells.

While this is an interesting finding, there are some points that should be addressed before this research article is consider for publication.

Major points:

1- In line 169 it is stated that almost all cells detached and underwent apoptosis within 48 hours (Fig. 2a, two images at bottom) after the use of the selective inhibitor of Oxidosqualene cyclase, although there are no assessments of apoptosis. Since the authors did not conduct a concentrations dependent experiment on Ro48-8071, toxicity of the compound cannot be excluded. It has been previously shown the effect of Ro48-8071 in (nM), therefore, I will recommend to perform a concentrations dependent experiment on Ro48-8071 with maximum concentration of 1 μM on cells cultured in DA-X condition. Authors may include the results in a supplementary document and revise the main results accordingly, if needed.

 

Minor points:

1- In line 106, reference is missing.

2- Correct the spelling of “Oxidosqualen cyclase” to Oxidosqualene cyclase.

3- Enhance the resolution of the histograms in figure 3b and 4b, so the entire information can be easily seen.

Reviewer #2: The manuscript "The amount range of membrane cholesterol required for robust cell adhesion and proliferation in serum-free condition by Takii et al. is well-studied and well-written. I recommend for publication.

Minor comments

1. The title is not conclusive and if "range" is removed, the title would be better received. or The title can be modified for better and as conclusive.

6. PLOS authors have the option to publish the peer review history of their article (what does this mean?). If published, this will include your full peer review and any attached files.

Reviewer #1: No

Reviewer #2: No

---

## [Author Response · Author response to Decision Letter 0]

11 Mar 2022

Dr. Avaniyapuram Kannan Murugan, M.Phil., Ph.D.

3rd March, 2022

Dear Dr. Murugan,

I would like to express my sincere gratitude to you for taking your valuable time to an evaluation and provide useful comments regarding our manuscript, “The amount of membrane cholesterol required for robust cell adhesion and proliferation in serum-free condition” (the title has been changed in response to reviewers’ comment). I would also like to thank the editor and reviewers for their constructive advice, which will greatly improve our paper. We agree with you that reason for the decision as minor revision, so we have addressed as below in response to “additional editor comments” and “Review Comments to the Author” by changing the content to better conform with the formatting and content rules of PLOS ONE.

I have separated my responses to the reviewers’ comments according to each reviewer.

Reviewer #1: 

While this is an interesting finding, there are some points that should be addressed before this re-search article is consider for publication.

Major points:

1- In line 169 it is stated that almost all cells detached and underwent apoptosis within 48 hours (Fig. 2a, two images at bottom) after the use of the selective inhibitor of Oxidosqualene cyclase, alt-hough there are no assessments of apoptosis. 

2- Since the authors did not conduct a concentrations dependent experiment on Ro48-8071, toxicity of the compound cannot be excluded. It has been previously shown the effect of Ro48-8071 in (nM), therefore, I will recommend to perform a concentrations dependent experiment on Ro48-8071 with maximum concentration of 1 μM on cells cultured in DA-X condition. Authors may include the re-sults in a supplementary document and revise the main results accordingly, if needed.

Our responses to Major points:

1- We fully agree with you and have incorporated this suggestion. In our revised manuscript, we de-scribed it as “cell death”, not “apoptosis” in line 171.

2- You have raised an important question regarding “toxicity of the compound”; since we’re sorry that this part was not clear in the original manuscript. However, we believe that it can be excluded from the conclusion since the harmful effects of Ro48-8071 even at 1µM on HeLa cells cultured in 10%FBS/DMEM were not shown at all in figure 2a (two images at the top). Therefore, we should have explained that “Ro48-8071 known as a cholesterol biosynthesis inhibition by selective inhibi-tion of Oxidosqualene cyclase had no harmful effects at all on adherent HeLa cells cultured in 10% FBS/ DMEM medium.” in line 167-169, as we have revised the contents of this part.

Minor points:

1- In line 106, reference is missing.

2- Correct the spelling of “Oxidosqualen cyclase” to Oxidosqualene cyclase.

3- Enhance the resolution of the histograms in figure 3b and 4b, so the entire information can be easily seen.

Our responses to Minor points:

1- We have incorporated your comments by the addition of an appropriate reference in line 106. In addition, we realized with your comment to need an appropriate reference regarding the function of Methyl-�-cyclodextrin and so made it in line 166.

2- We have corrected spelling of this throughout our revised paper (in line 106 and 168).

3- By checking the files of figure at submission, the resolution of the histograms in figure 3b and 4b was high enough to clearly see the information. This is probably due to the conversion of the image file format when the manuscript was compiled for sending to reviewers. Therefore, we believe that sufficient resolution of the histograms can be ensured for publication.

Reviewer #2: The manuscript "The amount range of membrane cholesterol required for robust cell adhesion and proliferation in serum-free condition by Takii et al. is well-studied and well-written. I recommend for publication.

Minor comments

1. The title is not conclusive and if "range" is removed, the title would be better received. or The title can be modified for better and as conclusive.

Our responses to Minor points:

1- We fully agree with you and have incorporated this suggestion by removing the word “range” from the title (in line 2). I would like to express our thanks for allowing us an opportunity to reconsider the title for more conclusive.

With these changes to our final manuscript, we hereby resubmit our manuscript for a secondary evalu-ation. We are certain that you will find this most recent version of our manuscript clears up the main issues you indicated in your response.

Thank you one again for your kind consideration.

Sincerely yours,

Daiji Okamura, Ph.D.

Department of Advanced Bioscience,

Graduate School of Agriculture,

Kindai University

TEL: +81-742-43-5384; FAX: +81-742-43-8976; e-mail: dokamura@nara.kindai.ac.jp

---

## [Decision Letter · Decision Letter 1]

12 Apr 2022

The amount of membrane cholesterol required for robust cell adhesion and proliferation in serum-free condition.

PONE-D-21-33428R1

Dear Dr. Okamura,

We’re pleased to inform you that your manuscript has been judged scientifically suitable for publication and will be formally accepted for publication once it meets all outstanding technical requirements.

Kind regards,

Avaniyapuram Kannan Murugan, M.Phil., Ph.D.

Academic Editor

PLOS ONE

Additional Editor Comments (optional):

Reviewers' comments:

Reviewer's Responses to Questions

**Comments to the Author**

1. If the authors have adequately addressed your comments raised in a previous round of review and you feel that this manuscript is now acceptable for publication, you may indicate that here to bypass the “Comments to the Author” section, enter your conflict of interest statement in the “Confidential to Editor” section, and submit your "Accept" recommendation.

Reviewer #1: All comments have been addressed

Reviewer #2: All comments have been addressed

2. Is the manuscript technically sound, and do the data support the conclusions?

Reviewer #1: Yes

Reviewer #2: Yes

3. Has the statistical analysis been performed appropriately and rigorously? 

Reviewer #1: Yes

Reviewer #2: Yes

4. Have the authors made all data underlying the findings in their manuscript fully available?

Reviewer #1: Yes

Reviewer #2: Yes

5. Is the manuscript presented in an intelligible fashion and written in standard English?

Reviewer #1: Yes

Reviewer #2: Yes

6. Review Comments to the Author

Reviewer #1: (No Response)

Reviewer #2: Manuscript: The amount of membrane cholesterol required for robust cell adhesion and proliferation in serum-free condition.

The authors have addressed all the comments. I do not have further comments.

7. PLOS authors have the option to publish the peer review history of their article (what does this mean?). If published, this will include your full peer review and any attached files.

Reviewer #1: No

Reviewer #2: No

---

## [Editor Report · Acceptance letter]

28 Jun 2022

PONE-D-21-33428R1 

The amount of membrane cholesterol required for robust cell adhesion and proliferation in serum-free condition. 

Dear Dr. Okamura:

I'm pleased to inform you that your manuscript has been deemed suitable for publication in PLOS ONE. Congratulations! Your manuscript is now with our production department. 

Kind regards, 

on behalf of

Dr. Avaniyapuram Kannan Murugan 

Academic Editor

PLOS ONE